# Label Distribution Learning Forests

**Wei Shen**[1,2]**, Kai Zhao**[1]**, Yilu Guo**[1]**, Alan Yuille**[2]
[1] Key Laboratory of Specialty Fiber Optics and Optical Access Networks,
Shanghai Institute for Advanced Communication and Data Science,
School of Communication and Information Engineering, Shanghai University
[2] Department of Computer Science, Johns Hopkins University
{shenwei1231,zhaok1206,gyl.luan0,alan.l.yuille}@gmail.com

## Abstract

Label distribution learning (LDL) is a general learning framework, which assigns to an instance a distribution over a set of labels rather than a single label or multiple labels. Current LDL methods have either restricted assumptions on the expression form of the label distribution or limitations in representation learning, e.g., to learn deep features in an end-to-end manner. This paper presents label distribution learning forests (LDLFs) - a novel label distribution learning algorithm based on differentiable decision trees, which have several advantages: 1) Decision trees have the potential to model any general form of label distributions by a mixture of leaf node predictions. 2) The learning of differentiable decision trees can be combined with representation learning. We define a distribution-based loss function for a forest, enabling all the trees to be learned jointly, and show that an update function for leaf node predictions, which guarantees a strict decrease of the loss function, can be derived by variational bounding. The effectiveness of the proposed LDLFs is verified on several LDL tasks and a computer vision application, showing significant improvements to the state-of-the-art LDL methods.

## 1 Introduction

Label distribution learning (LDL) [6, 11] is a learning framework to deal with problems of label ambiguity. Unlike single-label learning (SLL) and multi-label learning (MLL) [26], which assume an instance is assigned to a single label or multiple labels, LDL aims at learning the relative importance of each label involved in the description of an instance, i.e., a distribution over the set of labels. Such a learning strategy is suitable for many real-world problems, which have label ambiguity. An example is facial age estimation [8]. Even humans cannot predict the precise age from a single facial image. They may say that the person is probably in one age group and less likely to be in another. Hence it is more natural to assign a distribution of age labels to each facial image (Fig. 1(a)) instead of using a single age label. Another example is movie rating prediction [7]. Many famous movie review web sites, such as Netflix, IMDb and Douban, provide a crowd opinion for each movie specified by the distribution of ratings collected from their users (Fig. 1(b)). If a system could precisely predict such a rating distribution for every movie before it is released, movie producers can reduce their investment risk and the audience can better choose which movies to watch.

Many LDL methods assume the label distribution can be represented by a maximum entropy model [2] and learn it by optimizing an energy function based on the model [8, 11, 28, 6]. But, the exponential part of this model restricts the generality of the distribution form, e.g., it has difficulty in representing mixture distributions. Some other LDL methods extend the existing learning algorithms, e.g, by boosting and support vector regression, to deal with label distributions [7, 27], which avoid making this assumption, but have limitations in representation learning, e.g., they do not learn deep features in an end-to-end manner.

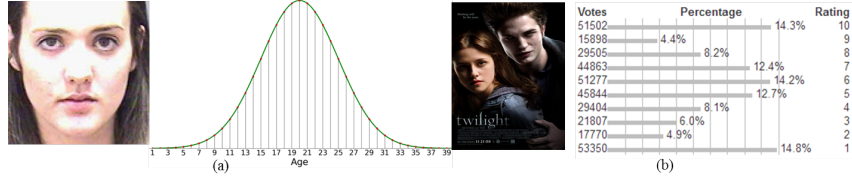

| Votes | Percentage | | Rating |
|---|---|---|---|
| 51502 | | 14.3% | 10 |
| 15898 | 4.4% | | 9 |
| 29505 | 8.2% | | 8 |
| 44863 | | 12.4% | 7 |
| 51277 | | 14.2% | 6 |
| 45844 | | 12.7% | 5 |
| 29404 | 8.1% | | 4 |
| 21807 | 6.0% | | 3 |
| 17770 | 4.9% | | 2 |
| 53350 | | 14.8% | 1 |

(a)        (b)

Figure 1: The real-world data which are suitable to be modeled by label distribution learning. (a) Estimated facial ages (a unimodal distribution). (b) Rating distribution of crowd opinion on a movie (a multimodal distribution).

In this paper, we present *label distribution learning forests* (LDLFs) - a novel label distribution learning algorithm inspired by differentiable decision trees [20]. Extending differentiable decision trees to deal with the LDL task has two advantages. One is that decision trees have the potential to model any general form of label distributions by mixture of the leaf node predictions, which avoid making strong assumption on the form of the label distributions. The second is that the split node parameters in differentiable decision trees can be learned by back-propagation, which enables a combination of tree learning and representation learning in an end-to-end manner. We define a distribution-based loss function for a tree by the Kullback-Leibler divergence (K-L) between the ground truth label distribution and the distribution predicted by the tree. By fixing split nodes, we show that the optimization of leaf node predictions to minimize the loss function of the tree can be addressed by *variational bounding* [19, 29], in which the original loss function to be minimized gets iteratively replaced by a decreasing sequence of upper bounds. Following this optimization strategy, we derive a discrete iterative function to update the leaf node predictions. To learn a forest, we average the losses of all the individual trees to be the loss for the forest and allow the split nodes from different trees to be connected to the same output unit of the feature learning function. In this way, the split node parameters of all the individual trees can be learned jointly. Our LDLFs can be used as a (shallow) stand-alone model, and can also be integrated with any deep networks, i.e., the feature learning function can be a linear transformation and a deep network, respectively. Fig. 2 illustrates a sketch chart of our LDLFs, where a forest consists of two trees is shown.

We verify the effectiveness of our model on several LDL tasks, such as crowd opinion prediction on movies and disease prediction based on human genes, as well as one computer vision application, i.e., facial age estimation, showing significant improvements to the state-of-the-art LDL methods. The label distributions for these tasks include both unimodal distributions (e.g., the age distribution in Fig. 1(a)) and mixture distributions (the rating distribution on a movie in Fig. 1(b)). The superiority of our model on both of them verifies its ability to model any general form of label distributions

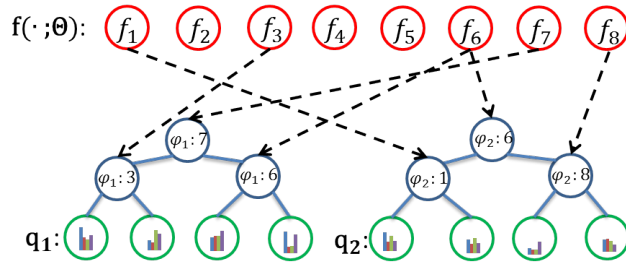

Figure 2: Illustration of a label distribution learning forest. The top circles denote the output units of the function $\mathbf{f}$ parameterized by $\Theta$, which can be a feature vector or a fully-connected layer of a deep network. The blue and green circles are split nodes and leaf nodes, respectively. Two index function $\varphi_1$ and $\varphi_2$ are assigned to these two trees respectively. The black dash arrows indicate the correspondence between the split nodes of these two trees and the output units of function $\mathbf{f}$. Note that, one output unit may correspond to the split nodes belonging to different trees. Each tree has independent leaf node predictions $\mathbf{q}$ (denoted by histograms in leaf nodes). The output of the forest is a mixture of the tree predictions. $\mathbf{f}(\cdot; \Theta)$ and $\mathbf{q}$ are learned jointly in an end-to-end manner.

## 2 Related Work

Since our LDL algorithm is inspired by differentiable decision trees, it is necessary to first review some typical techniques of decision trees. Then, we discuss current LDL methods.

**Decision trees**. Random forests or randomized decision trees [16, 1, 3, 4], are a popular ensemble predictive model suitable for many machine learning tasks. In the past, learning of a decision tree was based on heuristics such as a greedy algorithm where locally-optimal hard decisions are made at each split node [1], and thus, cannot be integrated into in a deep learning framework, i.e., be combined with representation learning in an end-to-end manner.

The newly proposed *deep neural decision forests* (dNDFs) [20] overcomes this problem by introducing a soft differentiable decision function at the split nodes and a global loss function defined on a tree. This ensures that the split node parameters can be learned by back-propagation and leaf node predictions can be updated by a discrete iterative function.

Our method extends dNDFs to address LDL problems, but this extension is non-trivial, because learning leaf node predictions is a constrained convex optimization problem. Although a step-size free update function was given in dNDFs to update leaf node predictions, it was only proved to converge for a classification loss. Consequently, it was unclear how to obtain such an update function for other losses. We observed, however, that the update function in dNDFs can be derived from *variational bounding*, which allows us to extend it to our LDL loss. In addition, the strategies used in LDLFs and dNDFs to learning the ensemble of multiple trees (forests) are different: 1) we explicitly define a loss function for forests, while only the loss function for a single tree was defined in dNDFs; 2) we allow the split nodes from different trees to be connected to the same output unit of the feature learning function, while dNDFs did not; 3) all trees in LDLFs can be learned jointly, while trees in dNDFs were learned alternatively. These changes in the ensemble learning are important, because as shown in our experiments (Sec. 4.4), LDLFs can get better results by using more trees, but by using the ensemble strategy proposed in dNDFs, the results of forests are even worse than those for a single tree.

To sum up, w.r.t. dNDFs [20], the contributions of LDLFs are: first, we extend from classification [20] to distribution learning by proposing a distribution-based loss for the forests and derive the gradient to learn splits nodes w.r.t. this loss; second, we derived the update function for leaf nodes by variational bounding (having observed that the update function in [20] was a special case of variational bounding); last but not the least, we propose above three strategies to learning the ensemble of multiple trees, which are different from [20], but we show are effective.

**Label distribution learning**. A number of specialized algorithms have been proposed to address the LDL task, and have shown their effectiveness in many computer vision applications, such as facial age estimation [8, 11, 28], expression recognition [30] and hand orientation estimation [10].

Geng *et al.* [8] defined the label distribution for an instance as a vector containing the probabilities of the instance having each label. They also gave a strategy to assign a proper label distribution to an instance with a single label, i.e., assigning a Gaussian or Triangle distribution whose peak is the single label, and proposed an algorithm called IIS-LLD, which is an iterative optimization process based on a two-layer energy based model. Yang *et al.* [28] then defined a three-layer energy based model, called SCE-LDL, in which the ability to perform feature learning is improved by adding the extra hidden layer and sparsity constraints are also incorporated to ameliorate the model. Geng [6] developed an accelerated version of IIS-LLD, called BFGS-LDL, by using quasi-Newton optimization. All the above LDL methods assume that the label distribution can be represented by a maximum entropy model [2], but the exponential part of this model restricts the generality of the distribution form.

Another way to address the LDL task, is to extend existing learning algorithms to deal with label distributions. Geng and Hou [7] proposed LDSVR, a LDL method by extending support vector regressor, which fit a sigmoid function to each component of the distribution simultaneously by a support vector machine. Xing *et al.* [27] then extended boosting to address the LDL task by additive weighted regressors. They showed that using the vector tree model as the weak regressor can lead to better performance and named this method AOSO-LDLLogitBoost. As the learning of this tree model is based on locally-optimal hard data partition functions at each split node, AOSO-LDLLogitBoost is unable to be combined with representation learning. Extending current deep learning algorithms to

address the LDL task is an interesting topic. But, the existing such a method, called DLDL [5], still focuses on maximum entropy model based LDL.

Our method, LDLFs, extends differentiable decision trees to address LDL tasks, in which the predicted label distribution for a sample can be expressed by a linear combination of the label distributions of the training data, and thus have no restrictions on the distributions (e.g., no requirement of the maximum entropy model). In addition, thanks to the introduction of differentiable decision functions, LDLFs can be combined with representation learning, e.g., to learn deep features in an end-to-end manner.

## 3  Label Distribution Learning Forests

A forest is an ensemble of decision trees. We first introduce how to learn a single decision tree by label distribution learning, then describe the learning of a forest.

### 3.1  Problem Formulation

Let $\mathcal{X} = \mathbb{R}^m$ denote the input space and $\mathcal{Y} = \{y_1, y_2, \ldots, y_C\}$ denote the complete set of labels, where $C$ is the number of possible label values. We consider a label distribution learning (LDL) problem, where for each input sample $\mathbf{x} \in \mathcal{X}$, there is a label distribution $\mathbf{d} = (d_{\mathbf{x}}^{y_1}, d_{\mathbf{x}}^{y_2}, \ldots, d_{\mathbf{x}}^{y_C})^\top \in \mathbb{R}^C$. Here $d_{\mathbf{x}}^{y_c}$ expresses the probability of the sample $\mathbf{x}$ having the $c$-th label $y_c$ and thus has the constraints that $d_{\mathbf{x}}^{y_c} \in [0, 1]$ and $\sum_{c=1}^C d_{\mathbf{x}}^{y_c} = 1$. The goal of the LDL problem is to learn a mapping function $\mathbf{g} : \mathbf{x} \to \mathbf{d}$ between an input sample $\mathbf{x}$ and its corresponding label distribution $\mathbf{d}$.

Here, we want to learn the mapping function $\mathbf{g}(\mathbf{x})$ by a decision tree based model $\mathcal{T}$. A decision tree consists of a set of split nodes $\mathcal{N}$ and a set of leaf nodes $\mathcal{L}$. Each split node $n \in \mathcal{N}$ defines a split function $s_n(\cdot; \boldsymbol{\Theta}) : \mathcal{X} \to [0, 1]$ parameterized by $\boldsymbol{\Theta}$ to determine whether a sample is sent to the left or right subtree. Each leaf node $\ell \in \mathcal{L}$ holds a distribution $\mathbf{q}_\ell = (q_{\ell_1}, q_{\ell_2}, \ldots, q_{\ell_C})^\top$ over $\mathcal{Y}$, i.e, $q_{\ell_c} \in [0, 1]$ and $\sum_{c=1}^C q_{\ell_c} = 1$. To build a differentiable decision tree, following [20], we use a probabilistic split function $s_n(\mathbf{x}; \boldsymbol{\Theta}) = \sigma(f_{\varphi(n)}(\mathbf{x}; \boldsymbol{\Theta}))$, where $\sigma(\cdot)$ is a sigmoid function, $\varphi(\cdot)$ is an index function to bring the $\varphi(n)$-th output of function $\mathbf{f}(\mathbf{x}; \boldsymbol{\Theta})$ in correspondence with split node $n$, and $\mathbf{f} : \mathbf{x} \to \mathbb{R}^M$ is a real-valued feature learning function depending on the sample $\mathbf{x}$ and the parameter $\boldsymbol{\Theta}$, and can take any form. For a simple form, it can be a linear transformation of $\mathbf{x}$, where $\boldsymbol{\Theta}$ is the transformation matrix; For a complex form, it can be a deep network to perform representation learning in an end-to-end manner, then $\boldsymbol{\Theta}$ is the network parameter. The correspondence between the split nodes and the output units of function $\mathbf{f}$, indicated by $\varphi(\cdot)$ that is randomly generated before tree learning, i.e., which output units from "$\mathbf{f}$" are used for constructing a tree is determined randomly. An example to demonstrate $\varphi(\cdot)$ is shown in Fig. 2. Then, the probability of the sample $\mathbf{x}$ falling into leaf node $\ell$ is given by

$$p(\ell|\mathbf{x}; \boldsymbol{\Theta}) = \prod_{n \in \mathcal{N}} s_n(\mathbf{x}; \boldsymbol{\Theta})^{\mathbf{1}(\ell \in \mathcal{L}_n^l)}(1 - s_n(\mathbf{x}; \boldsymbol{\Theta}))^{\mathbf{1}(\ell \in \mathcal{L}_n^r)}, \tag{1}$$

where $\mathbf{1}(\cdot)$ is an indicator function and $\mathcal{L}_n^l$ and $\mathcal{L}_n^r$ denote the sets of leaf nodes held by the left and right subtrees of node $n$, $\mathcal{T}_n^l$ and $\mathcal{T}_n^r$, respectively. The output of the tree $\mathcal{T}$ w.r.t. $\mathbf{x}$, i.e., the mapping function $g$, is defined by

$$\mathbf{g}(\mathbf{x}; \boldsymbol{\Theta}, \mathcal{T}) = \sum_{\ell \in \mathcal{L}} p(\ell|\mathbf{x}; \boldsymbol{\Theta})\mathbf{q}_\ell. \tag{2}$$

### 3.2  Tree Optimization

Given a training set $\mathcal{S} = \{(\mathbf{x}_i, \mathbf{d}_i)\}_{i=1}^N$, our goal is to learn a decision tree $\mathcal{T}$ described in Sec. 3.1 which can output a distribution $\mathbf{g}(\mathbf{x}_i; \boldsymbol{\Theta}, \mathcal{T})$ similar to $\mathbf{d}_i$ for each sample $\mathbf{x}_i$. To this end, a straightforward way is to minimize the Kullback-Leibler (K-L) divergence between each $\mathbf{g}(\mathbf{x}_i; \boldsymbol{\Theta}, \mathcal{T})$ and $\mathbf{d}_i$, or equivalently to minimize the following cross-entropy loss:

$$R(\mathbf{q}, \boldsymbol{\Theta}; \mathcal{S}) = -\frac{1}{N} \sum_{i=1}^N \sum_{c=1}^C d_{\mathbf{x}_i}^{y_c} \log(g_c(\mathbf{x}_i; \boldsymbol{\Theta}, \mathcal{T})) = -\frac{1}{N} \sum_{i=1}^N \sum_{c=1}^C d_{\mathbf{x}_i}^{y_c} \log\left(\sum_{\ell \in \mathcal{L}} p(\ell|\mathbf{x}_i; \boldsymbol{\Theta})q_{\ell_c}\right), \tag{3}$$

where $\mathbf{q}$ denote the distributions held by all the leaf nodes $\mathcal{L}$ and $g_c(\mathbf{x}_i; \mathbf{\Theta}, \mathcal{T})$ is the $c$-th output unit of $\mathbf{g}(\mathbf{x}_i; \mathbf{\Theta}, \mathcal{T})$. Learning the tree $\mathcal{T}$ requires the estimation of two parameters: 1) the split node parameter $\mathbf{\Theta}$ and 2) the distributions $\mathbf{q}$ held by the leaf nodes. The best parameters $(\mathbf{\Theta}^*, \mathbf{q}^*)$ are determined by

$$(\mathbf{\Theta}^*, \mathbf{q}^*) = \arg \min_{\mathbf{\Theta}, \mathbf{q}} R(\mathbf{q}, \mathbf{\Theta}; \mathcal{S}). \tag{4}$$

To solve Eqn. 4, we consider an alternating optimization strategy: First, we fix $\mathbf{q}$ and optimize $\mathbf{\Theta}$; Then, we fix $\mathbf{\Theta}$ and optimize $\mathbf{q}$. These two learning steps are alternatively performed, until convergence or a maximum number of iterations is reached (defined in the experiments).

### 3.2.1   Learning Split Nodes

In this section, we describe how to learn the parameter $\mathbf{\Theta}$ for split nodes, when the distributions held by the leaf nodes $\mathbf{q}$ are fixed. We compute the gradient of the loss $R(\mathbf{q}, \mathbf{\Theta}; \mathcal{S})$ w.r.t. $\mathbf{\Theta}$ by the chain rule:

$$\frac{\partial R(\mathbf{q}, \mathbf{\Theta}; \mathcal{S})}{\partial \mathbf{\Theta}} = \sum_{i=1}^{N} \sum_{n \in \mathcal{N}} \frac{\partial R(\mathbf{q}, \mathbf{\Theta}; \mathcal{S})}{\partial f_{\varphi(n)}(\mathbf{x}_i; \mathbf{\Theta})} \frac{\partial f_{\varphi(n)}(\mathbf{x}_i; \mathbf{\Theta})}{\partial \mathbf{\Theta}}, \tag{5}$$

where only the first term depends on the tree and the second term depends on the specific type of the function $f_{\varphi(n)}$. The first term is given by

$$\frac{\partial R(\mathbf{q}, \mathbf{\Theta}; \mathcal{S})}{\partial f_{\varphi(n)}(\mathbf{x}_i; \mathbf{\Theta})} = \frac{1}{N} \sum_{c=1}^{C} d_{\mathbf{x}_i}^{y_c} \left( s_n(\mathbf{x}_i; \mathbf{\Theta}) \frac{g_c(\mathbf{x}_i; \mathbf{\Theta}, \mathcal{T}_n^r)}{g_c(\mathbf{x}_i; \mathbf{\Theta}, \mathcal{T})} - \left(1 - s_n(\mathbf{x}_i; \mathbf{\Theta})\right) \frac{g_c(\mathbf{x}_i; \mathbf{\Theta}, \mathcal{T}_n^l)}{g_c(\mathbf{x}_i; \mathbf{\Theta}, \mathcal{T})} \right), \tag{6}$$

where $g_c(\mathbf{x}_i; \mathbf{\Theta}, \mathcal{T}_n^l) = \sum_{\ell \in \mathcal{L}_n^l} p(\ell|\mathbf{x}_i; \mathbf{\Theta}) q_{\ell_c}$ and $g^c(\mathbf{x}_i; \mathbf{\Theta}, \mathcal{T}_n^r) = \sum_{\ell \in \mathcal{L}_n^r} p(\ell|\mathbf{x}_i; \mathbf{\Theta}) q_{\ell_c}$. Note that, let $\mathcal{T}_n$ be the tree rooted at the node $n$, then we have $g_c(\mathbf{x}_i; \mathbf{\Theta}, \mathcal{T}_n) = g_c(\mathbf{x}_i; \mathbf{\Theta}, \mathcal{T}_n^l) + g_c(\mathbf{x}_i; \mathbf{\Theta}, \mathcal{T}_n^r)$. This means the gradient computation in Eqn. 6 can be started at the leaf nodes and carried out in a bottom up manner. Thus, the split node parameters can be learned by standard back-propagation.

### 3.2.2   Learning Leaf Nodes

Now, fixing the parameter $\mathbf{\Theta}$, we show how to learn the distributions held by the leaf nodes $\mathbf{q}$, which is a constrained optimization problem:

$$\min_{\mathbf{q}} R(\mathbf{q}, \mathbf{\Theta}; \mathcal{S}), \mathbf{s.t.}, \forall \ell, \sum_{c=1}^{C} q_{\ell_c} = 1. \tag{7}$$

Here, we propose to address this constrained convex optimization problem by *variational bounding* [19, 29], which leads to a step-size free and fast-converged update rule for $\mathbf{q}$. In *variational bounding*, an original objective function to be minimized gets replaced by its bound in an iterative manner. A upper bound for the loss function $R(\mathbf{q}, \mathbf{\Theta}; \mathcal{S})$ can be obtained by Jensen's inequality:

$$R(\mathbf{q}, \mathbf{\Theta}; \mathcal{S}) = -\frac{1}{N} \sum_{i=1}^{N} \sum_{c=1}^{C} d_{\mathbf{x}_i}^{y_c} \log \left( \sum_{\ell \in \mathcal{L}} p(\ell|\mathbf{x}_i; \mathbf{\Theta}) q_{\ell_c} \right)$$

$$\leq -\frac{1}{N} \sum_{i=1}^{N} \sum_{c=1}^{C} d_{\mathbf{x}_i}^{y_c} \sum_{\ell \in \mathcal{L}} \xi_\ell(\bar{q}_{\ell_c}, \mathbf{x}_i) \log \left( \frac{p(\ell|\mathbf{x}_i; \mathbf{\Theta}) q_{\ell_c}}{\xi_\ell(\bar{q}_{\ell_c}, \mathbf{x}_i)} \right), \tag{8}$$

where $\xi_\ell(q_{\ell_c}, \mathbf{x}_i) = \frac{p(\ell|\mathbf{x}_i; \mathbf{\Theta}) q_{\ell_c}}{g_c(\mathbf{x}_i; \mathbf{\Theta}, \mathcal{T})}$. We define

$$\phi(\mathbf{q}, \bar{\mathbf{q}}) = -\frac{1}{N} \sum_{i=1}^{N} \sum_{c=1}^{C} d_{\mathbf{x}_i}^{y_c} \sum_{\ell \in \mathcal{L}} \xi_\ell(\bar{q}_{\ell_c}, \mathbf{x}_i) \log \left( \frac{p(\ell|\mathbf{x}_i; \mathbf{\Theta}) q_{\ell_c}}{\xi_\ell(\bar{q}_{\ell_c}, \mathbf{x}_i)} \right). \tag{9}$$

Then $\phi(\mathbf{q}, \bar{\mathbf{q}})$ is an upper bound for $R(\mathbf{q}, \mathbf{\Theta}; \mathcal{S})$, which has the property that for any $\mathbf{q}$ and $\bar{\mathbf{q}}$, $\phi(\mathbf{q}, \bar{\mathbf{q}}) \geq R(\mathbf{q}, \mathbf{\Theta}; \mathcal{S})$, and $\phi(\mathbf{q}, \mathbf{q}) = R(\mathbf{q}, \mathbf{\Theta}; \mathcal{S})$. Assume that we are at a point $\mathbf{q}^{(t)}$ corresponding to the $t$-th iteration, then $\phi(\mathbf{q}, \mathbf{q}^{(t)})$ is an upper bound for $R(\mathbf{q}, \mathbf{\Theta}; \mathcal{S})$. In the next iteration, $\mathbf{q}^{(t+1)}$ is chosen such that $\phi(\mathbf{q}^{(t+1)}, \mathbf{q}) \leq R(\mathbf{q}^{(t)}, \mathbf{\Theta}; \mathcal{S})$, which implies $R(\mathbf{q}^{(t+1)}, \mathbf{\Theta}; \mathcal{S}) \leq R(\mathbf{q}^{(t)}, \mathbf{\Theta}; \mathcal{S})$.

Consequently, we can minimize $\phi(\mathbf{q}, \bar{\mathbf{q}})$ instead of $R(\mathbf{q}, \Theta; \mathcal{S})$ after ensuring that $R(\mathbf{q}^{(t)}, \Theta; \mathcal{S}) = \phi(\mathbf{q}^{(t)}, \bar{\mathbf{q}})$, i.e., $\bar{\mathbf{q}} = \mathbf{q}^{(t)}$. So we have

$$\mathbf{q}^{(t+1)} = \arg\min_{\mathbf{q}} \phi(\mathbf{q}, \mathbf{q}^{(t)}), \mathbf{s.t.}, \forall \ell, \sum_{c=1}^{C} q_{\ell_c} = 1, \tag{10}$$

which leads to minimizing the Lagrangian defined by

$$\varphi(\mathbf{q}, \mathbf{q}^{(t)}) = \phi(\mathbf{q}, \mathbf{q}^{(t)}) + \sum_{\ell \in \mathcal{L}} \lambda_\ell (\sum_{c=1}^{C} q_{\ell_c} - 1), \tag{11}$$

where $\lambda_\ell$ is the Lagrange multiplier. By setting $\frac{\partial \varphi(\mathbf{q}, \mathbf{q}^{(t)})}{\partial q_{\ell_c}} = 0$, we have

$$\lambda_\ell = \frac{1}{N} \sum_{i=1}^{N} \sum_{c=1}^{C} d_{\mathbf{x}_i}^{y_c} \xi_\ell(q_{\ell_c}^{(t)}, \mathbf{x}_i) \text{ and } q_{\ell_c}^{(t+1)} = \frac{\sum_{i=1}^{N} d_{\mathbf{x}_i}^{y_c} \xi_\ell(q_{\ell_c}^{(t)}, \mathbf{x}_i)}{\sum_{c=1}^{C} \sum_{i=1}^{N} d_{\mathbf{x}_i}^{y_c} \xi_\ell(q_{\ell_c}^{(t)}, \mathbf{x}_i)}. \tag{12}$$

Note that, $q_{\ell_c}^{(t+1)}$ satisfies that $q_{\ell_c}^{(t+1)} \in [0, 1]$ and $\sum_{c=1}^{C} q_{\ell_c}^{(t+1)} = 1$. Eqn. 12 is the update scheme for distributions held by the leaf nodes. The starting point $\mathbf{q}_\ell^{(0)}$ can be simply initialized by the uniform distribution: $q_{\ell_c}^{(0)} = \frac{1}{C}$.

### 3.3 Learning a Forest

A forest is an ensemble of decision trees $\mathcal{F} = \{\mathcal{T}_1, \ldots, \mathcal{T}_K\}$. In the training stage, all trees in the forest $\mathcal{F}$ use the same parameters $\Theta$ for feature learning function $\mathbf{f}(\cdot; \Theta)$ (but correspond to different output units of $\mathbf{f}$ assigned by $\varphi$, see Fig. 2), but each tree has independent leaf node predictions $\mathbf{q}$. The loss function for a forest is given by averaging the loss functions for all individual trees: $R_{\mathcal{F}} = \frac{1}{K} \sum_{k=1}^{K} R_{\mathcal{T}_k}$, where $R_{\mathcal{T}_k}$ is the loss function for tree $\mathcal{T}_k$ defined by Eqn. 3. To learn $\Theta$ by fixing the leaf node predictions $\mathbf{q}$ of all the trees in the forest $\mathcal{F}$, based on the derivation in Sec. 3.2 and referring to Fig. 2, we have

$$\frac{\partial R_{\mathcal{F}}}{\partial \Theta} = \frac{1}{K} \sum_{i=1}^{N} \sum_{k=1}^{K} \sum_{n \in \mathcal{N}_k} \frac{\partial R_{\mathcal{T}_k}}{\partial f_{\varphi_k(n)}(\mathbf{x}_i; \Theta)} \frac{\partial f_{\varphi_k(n)}(\mathbf{x}_i; \Theta)}{\partial \Theta}, \tag{13}$$

where $\mathcal{N}_k$ and $\varphi_k(\cdot)$ are the split node set and the index function of $\mathcal{T}_k$, respectively. Note that, the index function $\varphi_k(\cdot)$ for each tree is randomly assigned before tree learning, and thus split nodes correspond to a subset of output units of $\mathbf{f}$. This strategy is similar to the random subspace method [17], which increases the randomness in training to reduce the risk of overfitting.

As for $\mathbf{q}$, since each tree in the forest $\mathcal{F}$ has its own leaf node predictions $\mathbf{q}$, we can update them independently by Eqn. 12, given by $\Theta$. For implementational convenience, we do not conduct this update scheme on the whole dataset $\mathcal{S}$ but on a set of mini-batches $\mathcal{B}$. The training procedure of a LDLF is shown in Algorithm. 1.

---

**Algorithm 1** The training procedure of a LDLF.

---

**Require:** $\mathcal{S}$: training set, $n_B$: the number of mini-batches to update $\mathbf{q}$
  Initialize $\Theta$ randomly and $\mathbf{q}$ uniformly, set $\mathcal{B} = \{\emptyset\}$
  **while** Not converge **do**
    **while** $|\mathcal{B}| < n_B$ **do**
      Randomly select a mini-batch $B$ from $\mathcal{S}$
      Update $\Theta$ by computing gradient (Eqn. 13) on $B$
      $\mathcal{B} = \mathcal{B} \bigcup B$
    **end while**
    Update $\mathbf{q}$ by iterating Eqn. 12 on $\mathcal{B}$
    $\mathcal{B} = \{\emptyset\}$
  **end while**

---

In the testing stage, the output of the forest $\mathcal{F}$ is given by averaging the predictions from all the individual trees: $\mathbf{g}(\mathbf{x}; \Theta, \mathcal{F}) = \frac{1}{K} \sum_{k=1}^{K} \mathbf{g}(\mathbf{x}; \Theta, \mathcal{T}_k)$.

# 4 Experimental Results

Our realization of LDLFs is based on "Caffe" [18]. It is modular and implemented as a standard neural network layer. We can either use it as a shallow stand-alone model (sLDLFs) or integrate it with any deep networks (dLDLFs). We evaluate sLDLFs on different LDL tasks and compare it with other stand-alone LDL methods. As dLDLFs can be learned from raw image data in an end-to-end manner, we verify dLDLFs on a computer vision application, i.e., facial age estimation. The default settings for the parameters of our forests are: tree number (5), tree depth (7), output unit number of the feature learning function (64), iteration times to update leaf node predictions (20), the number of mini-batches to update leaf node predictions (100), maximum iteration (25000).

## 4.1 Comparison of sLDLFs to Stand-alone LDL Methods

We compare our shallow model sLDLFs with other state-of-the-art stand-alone LDL methods. For sLDLFs, the feature learning function $\mathbf{f}(\mathbf{x}, \mathbf{\Theta})$ is a linear transformation of $\mathbf{x}$, i.e., the $i$-th output unit $f_i(\mathbf{x}, \boldsymbol{\theta}_i) = \boldsymbol{\theta}_i^\top \mathbf{x}$, where $\boldsymbol{\theta}_i$ is the $i$-th column of the transformation matrix $\mathbf{\Theta}$. We used 3 popular LDL datasets in [6], `Movie`, `Human Gene` and `Natural Scene`[1]. The samples in these 3 datasets are represented by numerical descriptors, and the ground truths for them are the rating distributions of crowd opinion on movies, the diseases distributions related to human genes and label distributions on scenes, such as plant, sky and cloud, respectively. The label distributions of these 3 datasets are mixture distributions, such as the rating distribution shown in Fig. 1(b). Following [7, 27], we use 6 measures to evaluate the performances of LDL methods, which compute the average similarity/distance between the predicted rating distributions and the real rating distributions, including 4 distance measures (K-L, Euclidean, S$\phi$rensen, Squared $\chi^2$) and two similarity measures (Fidelity, Intersection).

We evaluate our shallow model sLDLFs on these 3 datasets and compare it with other state-of-the-art stand-alone LDL methods. The results of sLDLFs and the competitors are summarized in Table 1. For `Movie` we quote the results reported in [27], as the code of [27] is not publicly available. For the results of the others two, we run code that the authors had made available. In all case, following [27, 6], we split each dataset into 10 fixed folds and do standard ten-fold cross validation, which represents the result by "mean$\pm$standard deviation" and matters less how training and testing data get divided. As can be seen from Table 1, sLDLFs perform best on all of the six measures.

Table 1: Comparison results on three LDL datasets [6]. "↑" and "↓" indicate the larger and the smaller the better, respectively.

| Dataset | Method | K-L ↓ | Euclidean ↓ | S$\phi$rensen ↓ | Squared $\chi^2$ ↓ | Fidelity ↑ | Intersection ↑ |
|---|---|---|---|---|---|---|---|
| Movie | sLDLF (ours) | **0.073±0.005** | **0.133±0.003** | **0.130±0.003** | **0.070±0.004** | **0.981±0.001** | **0.870±0.003** |
| | AOSO-LDLogitBoost [27] | 0.086±0.004 | 0.155±0.003 | 0.152±0.003 | 0.084±0.003 | 0.978±0.001 | 0.848±0.003 |
| | LDLogitBoost [27] | 0.090±0.004 | 0.159±0.003 | 0.155±0.003 | 0.088±0.003 | 0.977±0.001 | 0.845±0.003 |
| | LDSVR [7] | 0.092±0.005 | 0.158±0.004 | 0.156±0.004 | 0.088±0.004 | 0.977±0.001 | 0.844±0.004 |
| | BFGS-LDL [6] | 0.099±0.004 | 0.167±0.004 | 0.164±0.004 | 0.096±0.004 | 0.974±0.001 | 0.836±0.003 |
| | IIS-LDL [11] | 0.129±0.007 | 0.187±0.004 | 0.183±0.004 | 0.120±0.005 | 0.967±0.001 | 0.817±0.004 |
| Human Gene | sLDLF (ours) | **0.228±0.006** | **0.085±0.002** | **0.212±0.002** | **0.179±0.004** | **0.948±0.001** | **0.788±0.002** |
| | LDSVR [7] | 0.245±0.019 | 0.099±0.005 | 0.229±0.015 | 0.189±0.021 | 0.940±0.006 | 0.771±0.015 |
| | BFGS-LDL [6] | 0.231±0.021 | 0.076±0.006 | 0.231±0.012 | 0.211±0.018 | 0.938±0.008 | 0.769±0.012 |
| | IIS-LDL [11] | 0.239±0.018 | 0.089±0.006 | 0.253±0.009 | 0.205±0.012 | 0.944±0.003 | 0.747±0.009 |
| Natural Scene | sLDLF (ours) | **0.534±0.013** | **0.317±0.014** | **0.336±0.010** | **0.448±0.017** | **0.824±0.008** | **0.664±0.010** |
| | LDSVR [7] | 0.852±0.023 | 0.511±0.021 | 0.492±0.016 | 0.595±0.026 | 0.813±0.008 | 0.509±0.016 |
| | BFGS-LDL [6] | 0.856±0.061 | 0.475±0.029 | 0.508±0.026 | 0.716±0.041 | 0.722±0.021 | 0.492±0.026 |
| | IIS-LDL [11] | 0.879±0.023 | 0.458±0.014 | 0.539±0.011 | 0.792±0.019 | 0.686±0.009 | 0.461±0.011 |

## 4.2 Evaluation of dLDLFs on Facial Age Estimation

In some literature [8, 11, 28, 15, 5], age estimation is formulated as a LDL problem. We conduct facial age estimation experiments on `Morph` [24], which contains more than 50,000 facial images from about 13,000 people of different races. Each facial image is annotated with a chronological age. To generate an age distribution for each face image, we follow the same strategy used in [8, 28, 5], which uses a Gaussian distribution whose mean is the chronological age of the face image (Fig. 1(a)). The predicted age for a face image is simply the age having the highest probability in the predicted

label distribution. The performance of age estimation is evaluated by the mean absolute error (MAE) between predicted ages and chronological ages. As the current state-of-the-art result on `Morph` is obtain by fine-tuning DLDL [5] on VGG-Face [23], we also build a dLDLF on VGG-Face, by replacing the softmax layer in VGGNet by a LDLF. Following [5], we do standard 10 ten-fold cross validation and the results are summarized in Table. 2, which shows dLDLF achieve the state-of-the-art performance on `Morph`. Note that, the significant performance gain between deep LDL models (DLDL and dLDLF) and non-deep LDL models (IIS-LDL, CPNN, BFGS-LDL) and the superiority of dLDLF compared with DLDL verifies the effectiveness of end-to-end learning and our tree-based model for LDL, respectively.

Table 2: MAE of age estimation comparison on `Morph` [24].

| Method | IIS-LDL [11] | CPNN [11] | BFGS-LDL [6] | DLDL+VGG-Face [5] | dLDLF+VGG-Face (ours) |
|--------|--------------|-----------|--------------|-------------------|------------------------|
| MAE | 5.67±0.15 | 4.87±0.31 | 3.94±0.05 | 2.42±0.01 | **2.24±0.02** |

As the distribution of gender and ethnicity is very unbalanced in `Morph`, many age estimation methods [13, 14, 15] are evaluated on a subset of `Morph`, called `Morph_Sub` for short, which consists of 20,160 selected facial images to avoid the influence of unbalanced distribution. The best performance reported on `Morph_Sub` is given by D2LDL [15], a data-dependent LDL method. As D2LDL used the output of the "fc7" layer in AlexNet [21] as the face image features, here we integrate a LDLF with AlexNet. Following the experiment setting used in D2LDL, we evaluate our dLDLF and the competitors, including both SLL and LDL based methods, under six different training set ratios (10% to 60%). All of the competitors are trained on the same deep features used by D2LDL. As can be seen from Table 3, our dLDLFs significantly outperform others for all training set ratios.

Note that, the generated age distributions are unimodal distributions and the label distributions used in Sec. 4.1 are mixture distributions. The proposed method LDLFs achieve the state-of-the-art results on both of them, which verifies that our model has the ability to model any general form of label distributions.

Figure 3: MAE of age estimation comparison on `Morph_Sub`.

| Method | Training set ratio | | | | | |
|--------|------|------|------|------|------|------|
| | 10% | 20% | 30% | 40% | 50% | 60% |
| AAS [22] | 4.9081 | 4.7616 | 4.6507 | 4.5553 | 4.4690 | 4.4061 |
| LARR [12] | 4.7501 | 4.6112 | 4.5131 | 4.4273 | 4.3500 | 4.2949 |
| IIS-ALDL [9] | 4.1791 | 4.1683 | 4.1228 | 4.1107 | 4.1024 | 4.0902 |
| D2LDL [15] | 4.1080 | 3.9857 | 3.9204 | 3.8712 | 3.8560 | 3.8385 |
| dLDLF (ours) | **3.8495** | **3.6220** | **3.3991** | **3.2401** | **3.1917** | **3.1224** |

## 4.3 Time Complexity

Let $h$ and $s_B$ be the tree depth and the batch size, respectively. Each tree has $2^{h-1} - 1$ split nodes and $2^{h-1}$ leaf nodes. Let $D = 2^{h-1} - 1$. For one tree and one sample, the complexity of a forward pass and a backward pass are $O(D + D + 1 \times C) = O(D \times C)$ and $O(D + 1 \times C + D \times C) = O(D \times C)$, respectively. So for $K$ trees and $n_B$ batches, the complexity of a forward and backward pass is $O(D \times C \times K \times n_B \times s_B)$. The complexity of an iteration to update leaf nodes are $O(n_B \times s_B \times K \times C \times D + 1) = O(D \times C \times K \times n_B \times s_B)$. Thus, the complexity for the training procedure (one epoch, $n_B$ batches) and the testing procedure (one sample) are $O(D \times C \times K \times n_B \times s_B)$ and $O(D \times C \times K)$, respectively. LDLFs are efficient: On `Morph_Sub` (12636 training images, 8424 testing images), our model only takes 5250s for training (25000 iterations) and 8s for testing all 8424 images.

## 4.4 Parameter Discussion

Now we discuss the influence of parameter settings on performance. We report the results of rating prediction on `Movie` (measured by K-L) and age estimation on `Morph_Sub` with 60% training set ratio (measured by MAE) for different parameter settings in this section.

**Tree number**. As a forest is an ensemble model, it is necessary to investigate how performances change by varying the tree number used in a forest. Note that, as we discussed in Sec. 2, the ensemble strategy to learn a forest proposed in dNDFs [20] is different from ours. Therefore, it is necessary to see which ensemble strategy is better to learn a forest. Towards this end, we replace our ensemble strategy in dLDLFs by the one used in dNDFs, and name this method dNDFs-LDL. The corresponding shallow model is named by sNDFs-LDL. We fix other parameters, i.e., tree depth and

output unit number of the feature learning function, as the default setting. As shown in Fig. 4 (a), our ensemble strategy can improve the performance by using more trees, while the one used in dNDFs even leads to a worse performance than one for a single tree.

Observed from Fig. 4, the performance of LDLFs can be improved by using more trees, but the improvement becomes increasingly smaller and smaller. Therefore, using much larger ensembles does not yield a big improvement (On `Movie`, the number of trees $K = 100$: K-L = 0.070 *vs* $K = 20$: K-L = 0.071). Note that, not all random forests based methods use a large number of trees, e.g., Shotton *et al.* [25] obtained very good pose estimation results from depth images by only 3 decision trees.

**Tree depth**. Tree depth is another important parameter for decision trees. In LDLFs, there is an implicit constraint between tree depth $h$ and output unit number of the feature learning function $\tau$: $\tau \geq 2^{h-1} - 1$. To discuss the influence of tree depth to the performance of dLDLFs, we set $\tau = 2^{h-1}$ and fix tree number $K = 1$, and the performance change by varying tree depth is shown in Fig. 4 (b). We see that the performance first improves then decreases with the increase of the tree depth. The reason is as the tree depth increases, the dimension of learned features increases exponentially, which greatly increases the training difficulty. So using much larger depths may lead to bad performance (On `Movie`, tree depth $h = 18$: K-L = 0.1162 *vs* $h = 9$: K-L = 0.0831).

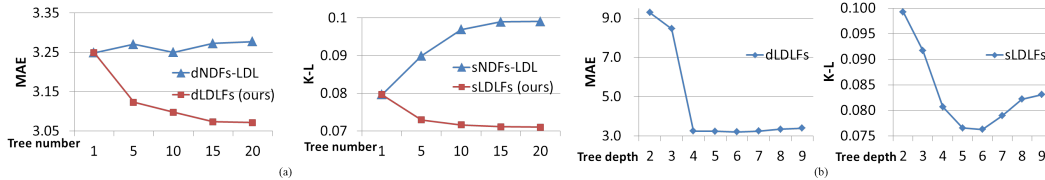

Figure 4: The performance change of age estimation on `Morph_Sub` and rating prediction on `Movie` by varying (a) tree number and (b) tree depth. Our approach (dLDLFs/sLDLFs) can improve the performance by using more trees, while using the ensemble strategy proposed in dNDFs (dNDFs-LDL/sNDFs-LDL) even leads to a worse performance than one for a single tree.

## 5    Conclusion

We present label distribution learning forests, a novel label distribution learning algorithm inspired by differentiable decision trees. We defined a distribution-based loss function for the forests and found that the leaf node predictions can be optimized via variational bounding, which enables all the trees and the feature they use to be learned jointly in an end-to-end manner. Experimental results showed the superiority of our algorithm for several LDL tasks and a related computer vision application, and verified our model has the ability to model any general form of label distributions.

**Acknowledgement**. This work was supported in part by the National Natural Science Foundation of China No. 61672336, in part by "Chen Guang" project supported by Shanghai Municipal Education Commission and Shanghai Education Development Foundation No. 15CG43 and in part by ONR N00014-15-1-2356.

## Footnotes

[1]We download these datasets from `http://cse.seu.edu.cn/people/xgeng/LDL/index.htm`.

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
