[Reviews · NeurIPS 2017]

Reviewer 1



The authors describe a method for label distribution learning based on differentiable decision trees. The authors use differentiable sigmoid units to estimate a label distribution using leaf nodes of trees. Learning in split nodes is done via backprop. whereas for leaf nodes the authors propose a methodology based on variational bounding. The authors compare their work with relevant methods on learning label distributions and show the competitiveness of their method. I think this is a good paper, providing a sound methodology for learning LD. The main weakness of this paper is that it is somewhat difficult for the reader to differentiate the actual contribution of this paper with respecto to reference [20]. I would like to see an answer to this in the rebuttal phase. detailed comments: typo: The best parameters ARE determined by: It is not clear to me whether the way split-nodes are learned is a contribution of this paper, or if it is described in [20], please clarify Do you have any plans of releasing the code of the LDF method with caffe? The numbers of trees and deep of trees is quite small, the authors make an effort to analyze the performance of their method under different parameter settings, but, the number of trees and depth is still small, what are author thoughts in much larger ensembles and tree depths? Have the authors performed experiments evaluating this setting? Please elaborate on this For the experimental results from Section 4.1, are the results from Table 1 comparable to each other? How do you guarantee this? Please provide more details For the problem approached in Section 4.2, can the authors please report the state of the art performance? (e.g., not only based on label distribution learning) What is the complexity of the model?, also, can the authors report runtime?

Reviewer 2



The paper describes a new label distribution learning algorithms based on differentiable decision trees. It can be seen as a natural extension to reference [20] (in the paper) where deep neural decision forests are introduced. To deal with label distribution learning problems the authors define an update function derive from variational bounding, define a loss function for forests, connect the split nodes from different trees to the same output, and proposed an approach where all the trees can be learned jointly. The authors followed an iterative process fixing alternatively the distribution at the leaf nodes and the split function. The authors tested their proposed method with three datasets and compared it against other state-of-the-art label distribution learning algorithms with very promising results. A limitation of the approach is that the number of trees in the ensemble and the depth of the trees need to be defined in advance. It also seems that the approach only works for binary trees, which again limits its applicability or require artificially deep trees. The authors should clarify how the index function works, which determines which output units from "f" are used for constructing the tree. The paper is well written, with clear results and sufficient contributions to be accepted at NIPS. Typo: - " ... update rule for q and ."